# Movement-goal relevant object shape properties act as poor but viable cues for the attribution of motor errors to external objects

**Shanaathanan Modchalingam**[1,2]ʘ\*, **Maria N. Ayala**[1,3]ʘ, **Denise Y. P. Henriques**[1,2]

**1** Centre for Vision Research, York University, Toronto, Ontario, Canada, **2** School of Kinesiology and Health Science, York University, Toronto, Ontario, Canada, **3** Department of Psychology, York University, Toronto, Ontario, Canada

ʘ These authors contributed equally to this work.
\* s.modcha@gmail.com

**Data Availability Statement:** All data and analysis scripts can be found in this project's Open Science Framework repository (https://osf.io/twgc8/).

## Abstract

When a context change is detected during motor learning, motor memories—internal models for executing movements within some context—may be created or existing motor memories may be activated and modified. Assigning credit to plausible causes of errors can allow for fast retrieval and activation of a motor memory, or a combination of motor memories, when the presence of such causes is detected. Features of the movement-context intrinsic to the movement dynamics, such as posture of the end effector, are often effective cues for detecting context change whereas features extrinsic to the movement dynamics, such as the colour of an object being moved, are often not. These extrinsic cues are typically not relevant to the motor task at hand and can be safely ignored by the motor system. We conducted two experiments testing if extrinsic but movement-goal relevant object-shape cues during an object-transport task can act as viable contextual cues for error assignment to the object, and the creation of new, object-shape-associated motor memories. In the first experiment we find that despite the object-shape cues, errors are primarily attributed to the hand transporting the object. In a second experiment, we find participants can execute differing movements cued by the object shape in a dual adaptation task, but the extent of adaptation is small, suggesting that movement-goal relevant object-shape properties are poor but viable cues for creating context specific motor memories.

## Introduction

Contextual information about the environment, the objects we interact with, and our own body is crucial for reliable motor control. Different environments may require different muscle activations to perform similar motor tasks, and changes to our end effectors, such as injuries to a hand or modifications to a tool, may require different movement kinematics. It is well established that the motor system can select from multiple previously-learned internal models, termed motor memories, cued by internal or external contexts to execute movements [1–6].

**Funding:** This work was supported by the Natural Science and Engineering Research Council of Canada Discovery Grant to DYPH and by the Natural Science and Engineering Research Council of Canada Post Graduate Scholarship - Doctoral to SM. The funders had no role in study design, data collection and analysis, decision to publish, or preparation of the manuscript.

**Competing interests:** The authors have declared that no competing interests exist.

Along with features of the movement itself [1, 7], contextual cues can aid in reliably selecting the correct internal model [8–11].

When the predicted consequences of a movement does not match the actual outcome, the difference, known as the sensory prediction error, can be used to update the internal models used to make the prediction [12–14]. Alternatively, if the sensory prediction error is sufficiently large, existing models that better predict the observed outcome given the intended movement may be activated. In the absence of any existing internal models that predict the observed outcome, new internal models may be created [1, 7].

The decision on whether to update an existing internal model or to create a new one can be informed by several factors. Features of the movement itself, as compared to predicted consequences of internal models, can be used to select an internal model for a given context [1]. Attributing errors to specific causes using available contextual cues can also allow for quick and reliable selection of internal models when those causes are detected [4, 5]. During tool use for example, errors can be attributed to external sources, such as the environment or the object being interacted with, and internal sources, such as the arm or hand used in the interaction. If the attribution of errors is accurate, models specific to the currently detected context can be accessed and updated. Additionally, accurate attribution of errors can allow for the switching of internal models before a movement needs to be initiated, and thus an error needs to be committed and perceived, allowing fast switching between multiple internal models and simultaneous adaptation to multiple contexts. In this study, we explore the types of cues that can lead to reliable attribution of errors to external sources and in turn facilitate fast switching of motor plans.

Cues that are "intrinsic" to a movement, i.e., cues that directly affect movement dynamics, such as movement direction, type, sequence, or posture, reliably facilitate fast context switching [9, 11, 15–19]. Findings on cues "extrinsic" to movement dynamics on the other hand, such as background colour and object identity are mixed [17, 19–23]. In general, extrinsic cues are often not sufficient for accurate error attribution during motor learning [20, 24]. Recent findings revealed that some extrinsic cues, such as colour [8], the action-effects of the effector [25], and visual location of the movement [25] can elicit dual adaptation if explicit adaptation processes (e.g., strategy formation) are engaged during adaptation, explaining some of the mixed findings and suggesting a critical role of cognitive strategy in dual adaptation [8]. Furthermore, extrinsic cues that inform of the dynamics of the interaction-point of tools [26, 27] can facilitate distinct motor memory formation. In these studies, the property of the tool that may cue the presence of a perturbation is task relevant. For example, in a study by McGarity-Shipley and colleagues [26], the location of the control point of a tool–the point with which the object interacts with the world–both cued the presence of opposing perturbations to movements and was integral to completing the task. We test if task relevancy itself can be enough of a cue to facilitate multiple motor memory formation.

In this study, we increased relevancy of an extrinsic cue (the shape of an object) to task goals by having participants perform an object transport task in an immersive virtual-reality environment where they move a 3D object to one of two receptacles. The shape of the object determined the receptacle it must be transported to. To isolate the object-shape cue, we applied a visuomotor rotation (VMR) to movements only when transporting objects (VMR direction mapped to object shape) while ensuring that movement dynamics, object dynamics and interaction dynamics were the same between differently shaped objects. In one experiment, we tested whether such an extrinsic cue was sufficient to allow for associated motor errors to be attributed to the object itself, rather than the hand. In a second experiment, we tested whether further increasing task relevancy by forcing attention to the object-shape in a dual adaptation task allowed for the formation of distinct motor memories for differently shaped objects. We

found that object-shape cues did not serve as sufficient contexts to elicit object-shape specific adaptation when adapting to a single visuomotor rotation, and were poor but viable cues for object-shape specific adaptation when adapting to multiple shape specific perturbations.

## Methods

### Participants

Sixty-two participants (47 Female, 19.87 ± 3.15 years of age, mean ± SD) participated in the study in one of two experiments: "Single Learning" (N = 32, 24 Female, 20.03 ± 3.21 years of age) and "Dual Learning" (N = 30, 23 Female, 19.69 ± 3.14 years of age). All participants were right-handed, had normal or corrected-to-normal vision, and were naïve to visuomotor learning experiments. Participation was voluntary and all participants provided written, informed consent. All data for this study was collected at the Sensorimotor Control Lab at York University between October 16, 2019, and March 03, 2020. The procedures used in this this study were approved by York University's Human Participant Review Committee and all experiments were performed in accordance with institutional and international guidelines.

### Apparatus

Participants sat on a height-adjustable chair facing a table placed at waist level. Participants donned a head-mounted display system (HMD: Oculus Rift Consumer Version 1; resolution 1080 by 1200 for each eye; refresh rate 90 Hz) and held an Oculus Touch controller in their right hand. Participants in the Single Learning experiments also held an Oculus Touch controller in their left hand. All visual feedback was provided via the HMD in an immersive virtual-reality experiment space developed in Unity 3D. We used the Unity Experiment Framework to handle trial and block schedules within the experiments [28]. Three Oculus Constellation sensors tracked the positions of the HMD and Oculus Touch controllers. To begin the experiments, participants placed their left hand on a magnetized docking apparatus (7.5 cm in diameter) attached to the table edge at body-midline. This apparatus acted as the starting position for each trial in the experiments and aided in returning the right hand to the starting position in the absence of visual feedback. The Oculus Touch controllers were equipped with magnets to aid attachment to the magnetized dock.

### Object-transport task

All visual feedback during the task was provided in the immersive virtual reality (VR) environment. In all experiments, participants repeatedly performed the Object-transport task. Each trial of the task began when participants returned the Oculus Touch controller held in their right hand to the magnetized dock. The dock was located near the starting position for each trial, termed the "Home" position. At the start of each trial, participants were shown an avatar-like representation of their hand (Fig 1A and 1B). An object in the shape of a cube (2 cm x 2 cm x 2 cm) or a sphere (2 cm diameter) appeared 10 cm away from the Home position, and two receptacles (box and cylinder shaped, depth of 2 cm) appeared 15 cm or 12 cm away from the object for the Single and Dual Learning experiments respectively. The opening of the box receptacle was 4 cm x 4cm x 4 cm and the opening of the cylinder was 4 cm in diameter. Participants were required to transport the object within 1 cm of the center of the receptacle in any orientation to complete the task. The target of the transport movement during a given Object-transport task was the receptacle with an opening matching the shape of the object.

 In the Single Learning experiment, participants transported objects of a single shape (counter-balanced between participants) during each block (Fig 1C). The trial-by-trial

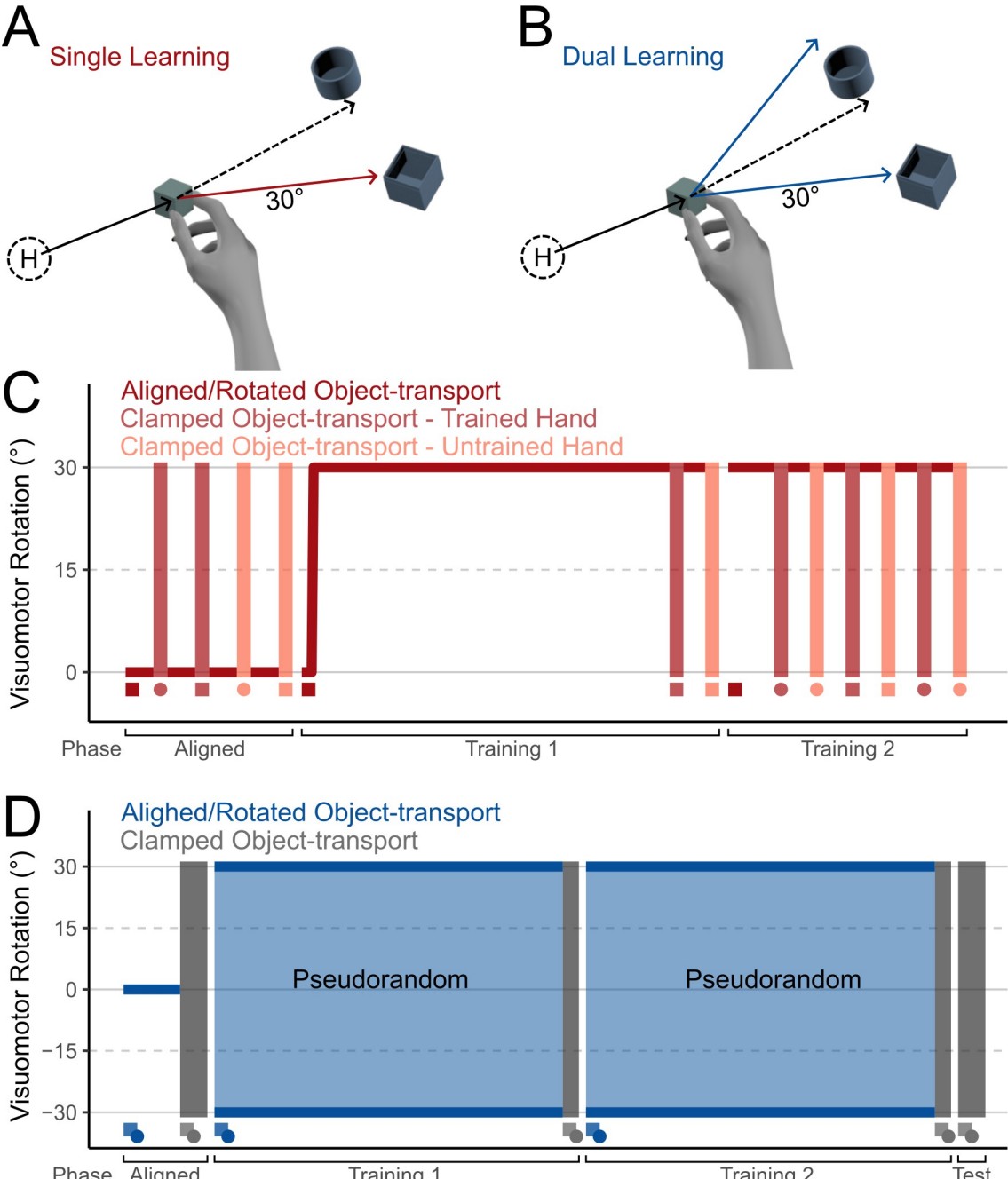

**Fig 1. Experiment setup and procedure.** A-B: Trial protocol during a Rotated Object-transport task in the Single (A) and Dual (B) Learning experiments. In both experiments, participants began a trial at the Home Position (H) and made a reaching movement toward an object. Each trial ended when participants transported the target to the corresponding receptacle. Solid lines represent visual movement direction and dashed lines represent real movement directions. C. The order of tasks during the Single Learning experiment. For each set of the Clamped Object-transport task, participants transported an object with the same or different shape as comparted to the object transported during the Rotated Object-transport tasks. D. The order of tasks during the Dual Learning experiment. All tasks in the Dual Learning experiment were performed using the right hand.

locations of the correct receptacles and the object's spawn locations were pseudorandomized, appearing at 45, 90, or 135 degrees in polar coordinates relative to the Home position. In the Dual Learning experiment, participants transported objects of both shapes during each block.

Here, the object-shapes appeared in trial-sets of 3, providing participants multiple opportunities to learn a shape-perturbation mapping before experiencing different perturbations to their movements. The trial-set-by-trial-set order of the object-shapes was pseudorandomized, ensuring that each object-shape appeared an equal number of times within a block. The trial-by-trial locations of the correct receptacles were pseudorandomized, appearing at 45, 90, or 135 degrees in polar coordinates centered on the object's spawn location.

Each trial contained two steps: Reach and Transport. During the Reach step, participants moved directly towards the object and made a pinch gesture using the index finger and thumb to interact with it. Once the gesture was completed the Reach step ended and the Transport step began. To highlight the object identity, the hand representation disappeared, and participants transported the object to the corresponding receptacle. A trial ended when participants moved the object to the centre of the corresponding receptacle.

In each experiment, participants performed three variations of the Object-transport task: Aligned, Rotated, and Clamped. The Reach step, during which the visual representation of the hand was spatially aligned with the participant's real hand, remained consistent across all variations. During the Transport step, the movement path of the object was aligned with the real movement path in the Aligned variation and a 30° visuomotor rotation (either clockwise or counterclockwise) was applied on the horizontal plane in the Rotated variation. In the Clamped variation, to prevent feedback-based corrections during a movement, the object's movement path was constrained to travel directly to the appropriate receptacle during the Transport step, with the object's distance from its starting position matched to the hand's distance from the object's initial location.

## Experiment protocol

All participants began the experiment with their right-hand controller magnetically attached to the magnetized dock. They were instructed to make smooth, direct movements throughout the experiment. Although no strict time limits were imposed, participants were encouraged by the experimenter to move in a quick and uniform pace throughout the experiment and were warned if they moved too slowly. Before beginning the experiments, participants completed practice trials to familiarize themselves with the virtual environment and task. Following the practice trials, participants completed one of either the Single Learning or Dual Learning experiments. Both experiments involved an Aligned phase, used to measure individual baseline performances, and one or more Training phases (Fig 1). The Dual Learning experiment also included a short Test phase. In all phases, participants repeatedly performed the Object-transport task. Participants performed the Aligned Object-transport task during the Aligned phase of the experiments and the Rotated Object-transport task during the Training phase. Participants also performed Clamped Object-transport tasks in all phases.

In the Single Learning experiment, we investigated if adapting to a perturbation when transporting an object could be linked to the object's shape if the shape was relevant to the task goal. Participants completed 3 phases in the experiment during a single session: The Aligned phase, and two Training phases (Fig 1C). To prevent fatigue, each phase of the experiment was preceded by a short break lasting a minimum of 30 seconds. During the Aligned phase, participants performed 60 trials of the Aligned Object-transport task using their dominant hand. Four 6-trial blocks of the Clamped Object-transport tasks were interspersed within the Aligned phase following each set of 15 Aligned trials. During the first Training phase, following 6 trials of the Aligned Object-transport task, participants adapted to a 30° visuomotor rotation (clockwise or counterclockwise, counter-balanced across participants) using a designated object-shape (either cube or sphere, counter-balanced across participants) for 180 trials of the Object-

transport task. Following the first set of Rotated Object-transport tasks, we tested if adaptation transferred to using either a differently shaped object or a different hand using Clamped Object-transport tasks where participants transported differently shaped objects, used a different hand to transport the object, or both. If adaptation was specific to the object shape, it would persist when participants switched hands, but diminish when they encountered a different object shape. Participants first performed two 6-trial blocks of the Clamped Object-transport task, with a 15-trial block of the Rotated Object-transport task in between. Then, in the second Training Phase, participants first performed 24 trials of the Rotated Object-transport task using the designated object-shape, then performed six 6-trial blocks of the Clamped Object-transport task, with 15-trial blocks of the Rotated Object-transport task in between.

In the Dual Learning experiment, we investigated whether participants could concurrently adapt to two distinct visuomotor rotations at the same time when the perturbations were matched to specific object-shapes. Here, participants completed 4 phases in the experiment during a single session: the Aligned phase, followed by two Training phases and a Test phase. Each phase was again preceded by a short break lasting a minimum of 30 seconds. During the Aligned phase, participants performed 30 trials of the Aligned Object-transport task. During each trial, participants transported one of the two possible object-shapes (cube or sphere, order pseudorandomized using 3-trial sets to ensure each shape was transported an equal number of times). During both Training phases, participants adapted to two opposing 30˚ visuomotor rotations mapped to two distinct object-shapes (cube or sphere, object-shape-rotation mappings counterbalanced across participants, order pseudorandomized using 3-trial sets to ensure each shape was transported an equal number of times) for 180 trials. We tested whether participants dually adapted to both visuomotor perturbations during the Training phase and tested whether such adaptation relied on feedback-based corrections using Clamped Object-transport tasks. Following the 180-trial block of Rotated Object-transport tasks, participants performed 6 trials of the Clamped Object-transport task (object-shape order pseudo-randomized using 3-trial sets). Additionally, to ascertain if participants experienced implicit dual adaptation, we conducted a short Test phase using 12 trials of the Clamped Object-transport task (object-shape order pseudo-randomized using 3-trial sets). To determine if participants adapted implicitly, we used an altered version of a process dissociation procedure in which we instructed participants to refrain from using any strategies they may have developed during the training phase [29, 30].

## Data analysis

All position measures were recorded using Unity 3D (2020.1.17, Unity Technologies, San Francisco, CA). Data was analyzed offline using R and was accessed for analysis from the start of data collection (October of 2019) to January of 2024. All data and analysis scripts, along with supplementary analysis notebooks, can be found in this project's Open Science Framework repository (https://osf.io/twgc8/).

We assessed performance in the object-transport task by calculating the hand angle (HA) of a movement. We defined HA as the angular deviation from a straight-line movement to the target at 3 cm from the object's starting position. HA is a measure of initial directional error, prior to effects of feedback-based corrections. We corrected for individual biases in transport behavior by subtracting individual baseline performances per Target location from the HA. Further, to compare across counterbalanced rotation directions, we normalized HA relative to the direction of the applied visuomotor rotation. We conducted all subsequent calculations and data analysis on the baseline corrected and normalized data.

To investigate performance changes over time, we compared mean HAs during the initial and final trial-sets of the first Training phase in each experiment, when participants were

abruptly introduced to the visuomotor perturbations. The initial trial-set consisted of 3 trials: one reach to each of the 3 available targets. The final trial-set consisted of 9 trials.

To determine if participants in the Single Learning experiment adapted to the visuomotor rotation, we compared HA during the Reach and Transport step of the first and last trial-sets during the Training phase using separate paired sample t-tests. We then tested possible error attribution to either the object or the hand by comparing HA during the Clamped Object-transport task under various conditions. We performed a 2x2 repeated-measures analysis of variance (ANOVA) with the hand (same or different hand) and the object shape (same or different object shape) as within-subject factors. We used Tukey's HSD to conduct pairwise analysis when post-hoc tests were necessary. For all statistical tests, we additionally computed Inclusion Bayes factors between models that include or do not include relevant effects [31].

To determine if participants in the Dual Learning experiment adapted to opposing rotations, we first determined if participants HAs changed over the course of the Training phase using paired sample t-tests. We separately compared the changes in HAs during the Reach and Transport steps. Additionally, to determine if participants performed differently without feedback-based corrections, we compared non-normalized HAs during the Transport step of Clamped Object-transport tasks in the Dual Learning experiment, where participants transported objects associated with positive and negative rotations using a paired sample t-test. To determine if object-shape related changes in HA were implicit, we compared non-normalized HAs during the Transport step of the Clamped Object-transport tasks when participants were instructed to not employ any strategies they may have learned to compensate for the visuomotor perturbations during the Training phase.

Finally, we tested whether the time taken to initiate and execute the Reach step, the time taken to initiate the Transport step, or the time taken to execute the Transport step changed during the Training phase of both experiments. For all three measures, we conducted separate 2x2 mixed ANOVAs with the experiment as a between-subject factor and the trial-set as a within-subject factor. When post-hoc tests were necessary, we used Tukey's HSD to perform pairwise comparisons.

## Results

We conducted two experiments to determine if goal-relevant visual properties of objects would suffice to enable error attribution to an object during Object-transport tasks. In the first experiment, participants transported an object with their right hand. A 30˚ visuomotor rotation was applied to their movement while transporting the object. To test for attribution of error, they were asked to transport a novel object, with either the trained hand or with their untrained, contralateral hand.

In the second experiment, to understand whether the goal-relevant extrinsic cues could facilitate dual adaptation, we mapped opposing visuomotor rotations (30˚ CCW or 30˚ CW) to distinct object shapes (sphere or cube). To test for dual adaptation, participants preformed Clamped Object-transport tasks where the direction of movement was clamped to head directly to the target. To test for possible implicit dual adaptation, participants also performed Clamped Object-transport tasks while disengaging any strategies they may have formed during the Training phase.

### Experiment 1: Single learning

**Visuomotor adaptation.** We compared normalized hand angles (HA) during the initial and final trial-sets of the Training phase (Fig 2A–2C). We found no evidence of change in the HAs during the Reach step, where the location of the hand was aligned with the real hand,

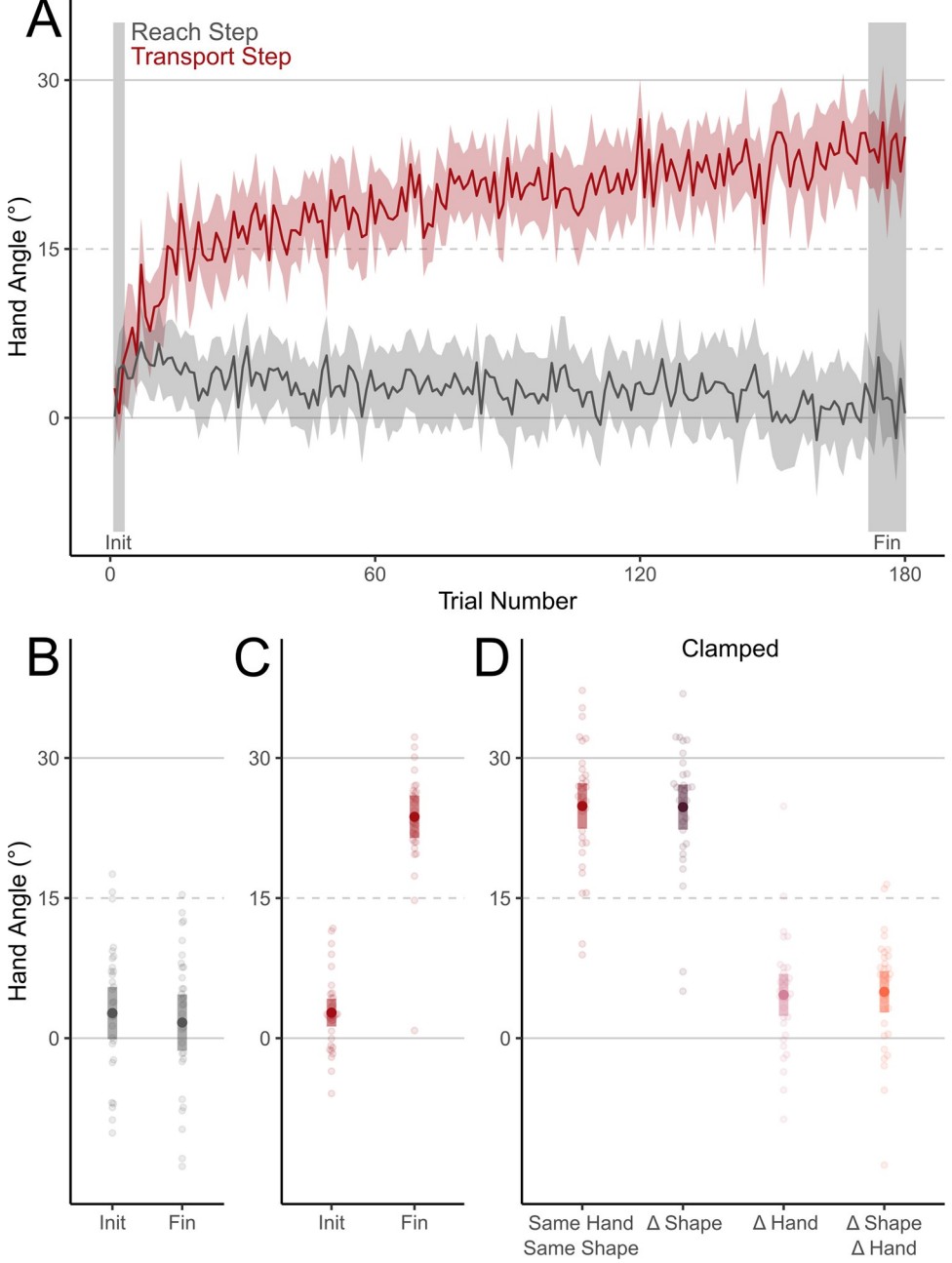

**Fig 2. Hand angles: Single Learning experiment.** Angular deviations from a straight-to-target movements (Hand Angle) during the Training phase in the Single Learning experiment. A. Hand Angles over the first 180 trials in the Training phase. B. Hand Angles during the Reach step of the initial and final trial-sets during the Training phase. C. Hand Angles during the Transport step of the initial and final trial-sets during the Training phase. D. Hand Angles during Clamped Object-transport tasks while participants transported objects of a different shape from the Rotated Object-transport task ($\Delta$ Shape), with a different hand from the one used in the Rotated Object-transport task ($\Delta$ Hand), and both. Shaded areas represent 95% CIs.

between the initial and final trial-sets of the Training phase ($t_{(31)}$ = -0.554, p = 0.583, $BF_{10}$ = 0.22). During the Transport step of the Object-transport task, participants adapted their movements to counter the visuomotor perturbation over the course of the Training phase ($t_{(31)}$ =

17.987, p < 0.001, $BF_{10}$ = 8.36x10$^{14}$, Fig 2B). Over 180 trials (60 trial-sets), participants adapted to 79.0% of the rotation (23.7˚ hand angle during the final trial-set of the Training phase).

**Error attribution.** We first determined if retrieval of the adapted motor memory was cued by the object-shape. After adapting to a 30˚ visuomotor rotation, we tested HAs during the Transport step of Clamped Object-transport tasks. (Fig 2D: main effect of hand: $F_{(1, 31)}$ = 174.588, p < 0.001, $\eta^2$ = 0.70, $BF_{incl}$ = 3.10x10$^{36}$). We found moderate evidence that the object shape did not affect HA ($F_{(1, 31)}$ = 0.030, p = 0.863, $\eta^2$ = 0.00007, $BF_{incl}$ = 0.152).

After collapsing across object shape, we found moderate evidence that participants performed the same as during the Rotated Object-transport tasks when transporting objects with the trained hand ($t_{(31)}$ = -0.588, p = 0.560, $BF_{10}$ = 0.22). Participants also retained a small but significant portion (~19%) of the change in HA developed during the Transport step of the Object-transport task when performing with the untrained hand ($t_{(31)}$ = 4.7836, p < 0.001, $BF_{10}$ = 589.64).

## Experiment 2: Dual learning

**Visuomotor adaptation.** Participants in the Dual Learning experiment completed a typical dual adaptation paradigm with object identity distinguishing the perturbation experienced on a given training trial. Overall, when collapsing across rotations, we found moderate evidence of no change in HA across the Training phase during the Reach step of the Object-transport task, where the location of the hand was aligned with the real hand ($t_{(29)}$ = 0.702, p = 0.488, $BF_{10}$ = 0.24). During the Transport step, participants were able to successfully counter 11.9% (3.59˚) of the 30˚ visuomotor rotation by the final 3 trial-sets of the Training phase relative to their HAs during the Baseline phase ($t_{(29)}$ = 3.68, p < 0.001, $BF_{10}$ = 35.0). We failed to find any evidence for changes in HAs during the Transport step across the Training phase ($t_{(29)}$ = 1.90, p = 0.067, $BF_{10}$ = 0.95), suggesting learning was approaching saturation within the first three trials of experiencing a perturbation (Fig 3A–3C).

For the Dual Learning experiment, we observed small changes in non-normalized HAs in the expected directions for each distinctive object during the Clamped Object-transport trials (Fig 3D). Participants' HAs differed between the two object shapes ($t_{(29)}$ = -2.58, p = 0.0153, $BF_{10}$ = 3.14). The magnitudes of the changes when compared to baseline performance were small: 1.47˚ and 3. 56˚ when transporting the shape associated with the clockwise and counterclockwise visuomotor rotation respectively (Fig 3C). These changes in non-normalized HA further decreased when participants were instructed not to employ a strategy, resulting in a lack of statistical reliability (Fig 3E: $t_{(29)}$ = -1.89, p = 0.069, $BF_{10}$ = 0.93). Given the large inter-subject variability and small magnitude of error, we do not make strong conclusions about implicit and explicit subprocesses underlying the observed dual adaptation.

## Reaction and execution times

We also tested if the time taken to initiate and execute the Reach step, the time taken to initiate the Transport step, or the time taken to execute the Transport step changed during the Training phase of both experiments (Fig 4). Although there was an interaction between the trial-set and experiment in time taken to complete the Reach step ($F_{(1, 60)}$ = 8.18, p = 0.006, $\eta^2$ = 0.062, $BF_{incl}$ = 21.76), this was driven by higher step-completion times in only the initial trial-set of Dual Learning experiment when compared to the final trial-sets of both the Single Learning (Tukey's HSD: $t_{(60)}$ = 3.62, p = 0.003), and Dual Learning (Tukey's HSD: $t_{(60)}$ = 4.29, p < 0.001) experiments (Fig 4A). By the end of the Training phase, participants in both experiments took a similar amount of time to complete the Reach step (Tukey's HSD: $t_{(60)}$ = -2.05,

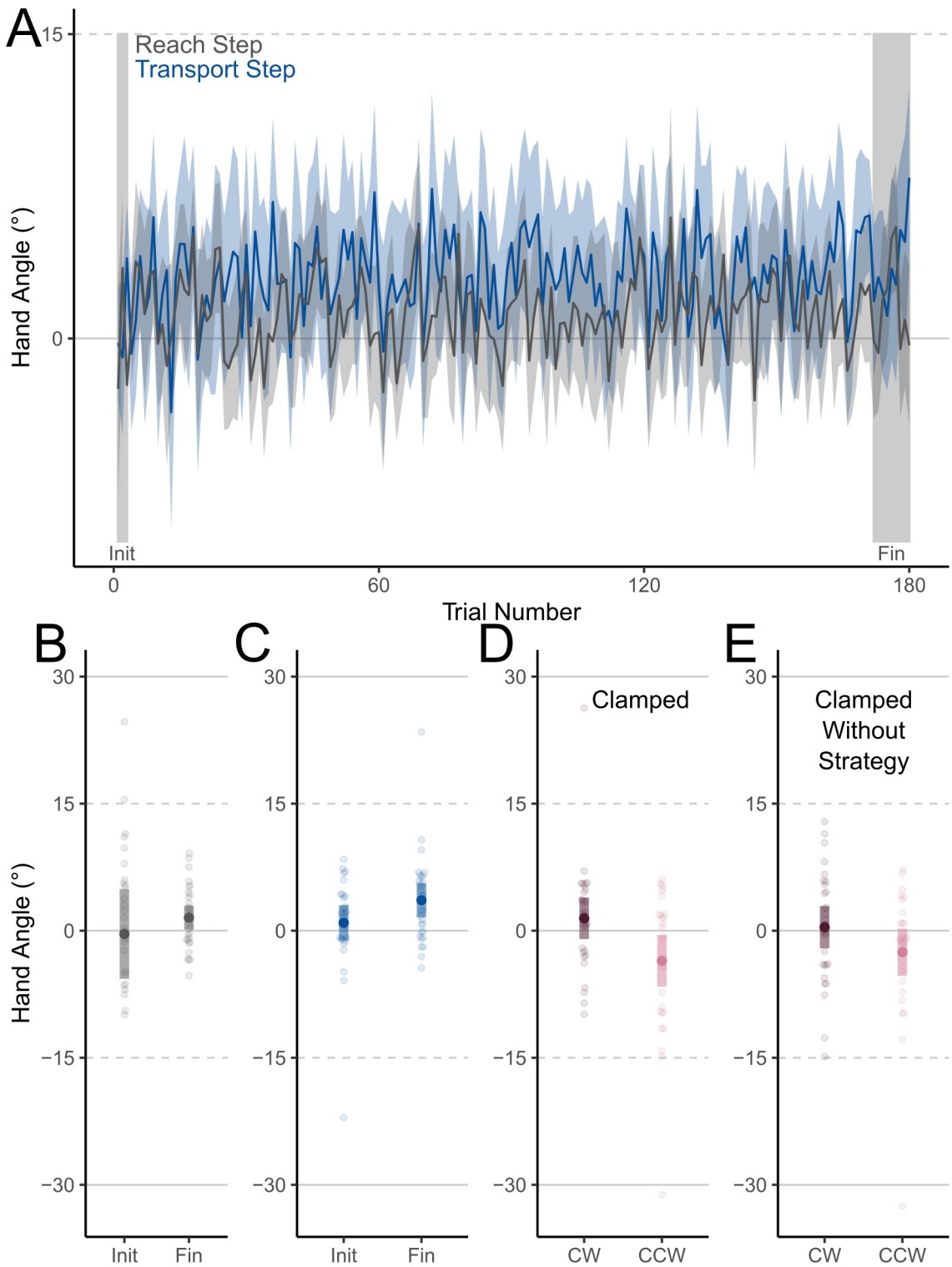

**Fig 3. Hand angles: Dual Learning experiment.** Angular deviations from a straight-to-target movements (Hand Angle) during the Training phase in the Dual Learning experiment. A. Hand Angles over the first 180 trials in the Training phase. B. Hand Angles during the Reach step of the initial and final trial-sets during the Training phase. C. Hand Angles during the Transport step of the initial and final trial-sets during the Training phase. D. Non-normalized Hand Angles during the Transport step of Clamped Object-transport tasks while transporting the object associated with clockwise (CW) and counterclockwise (CCW) perturbations. E. Non-normalized Hand Angles during the Transport step of Clamped Object-transport tasks while transporting the object associated with

clockwise (CW) and counterclockwise (CCW) perturbations when participants were instructed to exclude strategies that may have formed during the Training phase. Shaded areas represent 95% CIs.

p = 0.182). This may be due to the confusing nature of abruptly experiencing opposing perturbations.

Both experiment ($F_{(1, 60)} = 6.45$, p = 0.014, $\eta^2 = 0.060$, $BF_{incl} = 2.74$) and trial-set ($F_{(1, 60)} = 38.47$, p < 0.001, $\eta^2 = 0.205$, $BF_{incl} = 1.12 \times 10^6$) affected reaction times during the Transport step across the Training phase (Fig 4B). Surprisingly, participants in the Dual Learning group

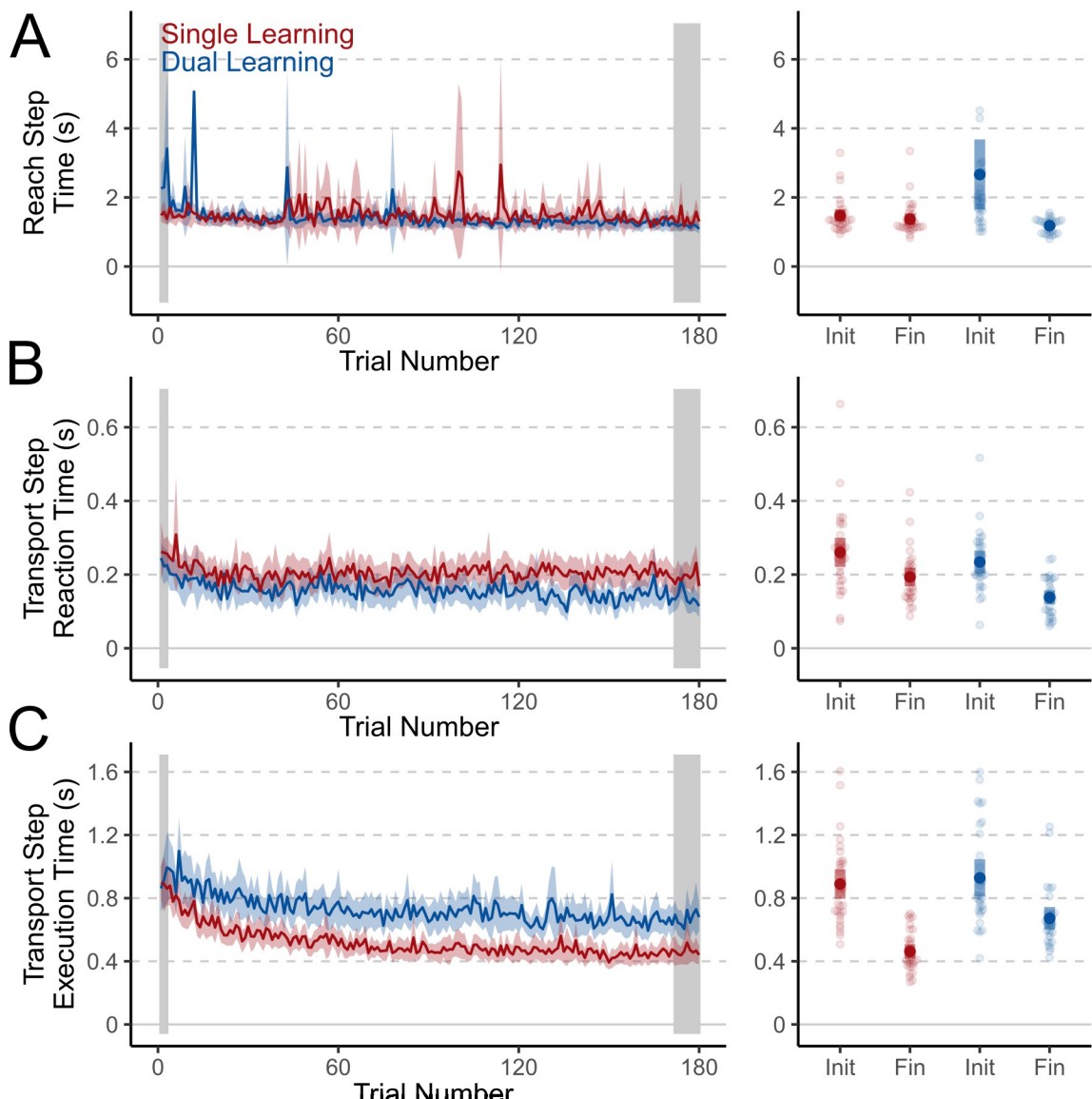

**Fig 4. Reach and Transport step movement times and reaction times.** Changes in movement initiation and execution times during Object-transport tasks during the Training phase of the Single Learning (red) and Dual Learning (blue) experiments. A. Time required to initiate and execute the Reach Step (Step 1) of the Object-transport task during the Training phase. B. Time to initiate movements during the Transport Step (Step 2) of the Object-transport task during the Training phase. C. Time required to execute the Transport Step (Step 2) once a movement had been initiated. Shaded areas represent 95% CIs.

were faster to start executing their transport movements by the end of training when compared to both the final trial-set of the Single Learning experiment (Tukey's HSD: $t_{(60)}$ = -3.52, p = 0.005) and the initial trial-set of the Dual Learning experiment (Tukey's HSD: $t_{(60)}$ = 5.08, p < 0.001). This may indicate low cognitive strategy use, leading to the small behavioural changes seen in the Rotated Object-transport task analysis.

Finally, the experiments differently affected execution time changes between initial and final trial-sets (Fig 4C: $F_{(1, 60)}$ = 4.95, p = 0.030, $\eta^2$ = 0.033, $BF_{incl}$ = 7.42). Despite similar execution times for the Transport step in both groups during the initial trial-set (Tukey's HSD: $t_{(60)}$ = 0.543, p = 0.948), and participants in the Dual Learning group having to transport objects a shorter distance than those in the Single Learning group, execution times were higher in the Dual Learning group in the final trial-set (Tukey's HSD: $t_{(60)}$ = 5.154, p < 0.001). This is likely due to additional correction times needed by participants making larger errors in the final trial-set of the Dual Learning experiment.

## Discussion

The formation of new motor memories associated with detectable contexts enables quick switching between motor contexts by activating associated motor plans when a specific context is detected. Assigning errors from movements to plausible causes is crucial for associating motor memories with specific contexts, as these causes serve as contextual cues for future retrieval of motor commands. Properties of movement end effectors, such as human body parts or external objects, that are intrinsic to movement dynamics reliably aid in forming motor memories tied to specific contexts. The role of properties extrinsic to movement dynamics, like visual properties of the object of interaction, is less well understood. Our findings show that visual object-shape cues linked to a visuomotor perturbation are insufficient for creating distinct motor memories when adapting to a single visuomotor rotation, even when the object-shape was relevant to task goals. When the mapping between two distinct object-shapes and two distinct perturbations were repeatedly experienced in a dual-learning paradigm, participants' performance differed between object-shapes, but expressed only partial adaptation.

In the Single Learning experiment, we directly tested if motor control changes to a visuomotor perturbation were linked to an object's shape. Participants adapted to a visuomotor perturbation with a single object-shape, and we assessed if the adaptation persisted when the objectshape or the hand used was changed. In our experiment learning was entirely attributed to the hand and environment, not the object-shape. When using the same hand for differently shaped objects, the adaptation effects transferred fully to the new object. This result suggests that object shape was not a strong enough contextual cue to facilitate assignment of motor error. This is a common finding for cues extrinsic to movement dynamics [20, 24]. While some studies show that purely visual properties of a task, such as colour [22], or additional movement of a visual end-effector [26] can facilitate distinct motor memory formation, our findings indicate that movement-goal relevant object-shape properties alone are likely insufficient.

When using the non-adapted hand to move objects, both familiar and unfamiliar, ~19% of the total adaptation was retained. This component of learning is consistent with previous findings [32], and may reflect learning to perform the task itself or updates to a baseline internal model for performing object-transport movements [1].

When using the adapted hand, performance during clamped trials in the Single Learning experiment, where participants were not asked to exclude strategies, was similar to performance during late training. Although decay of adaptation is minimized in reaching tasks with

clamped feedback, we did not expect complete retention from the rotated-feedback Object-transport tasks [33–36]. In our experiments, trials with clamped feedback and preceding trials with rotated feedback shared the same target locations requiring similar movement vectors to complete the task. In addition to model-based motor learning, model free mechanisms such as reinforcement learning to favour movement directions towards the learned target directions may have resulted in the apparent high retention and low decay of adaptation in our tasks with clamped feedback [37, 38].

Adaptation to errors attributed to objects has been observed in the past [32]. In a study testing the assignment of credit to errors during Object-transport tasks, Kong et al. [32] trained participants to transport familiar objects (cups). The researchers observed a reduction in measured adaptation when participants transported a different object while still using the trained hand. Since transporting cups is a common task, participants likely employed pre-existing internal models for this category of interactions that could be flexibly switched to given changes in object identity. Unlike previous studies, our study's virtual reality environment, and the dynamics of interactions within that environment, was unfamiliar to most participants. Since participants transported objects with both the left and the right hand during the Aligned phase of the experiment, and since posture, perspective, and the used hand can lead to the formation of distinct motor memories and enable dual adaptation [9, 25, 39], new hand-specific internal models for object transport may have been created, leading to low transfer across hands. The additional haptic feedback from transported real-world objects could further facilitate the development of motor memories specific to the movements involved in transporting these objects. Cues intrinsic to movements such as the altered forces or postures required when transporting real-world objects can reliably facilitate object-specific adaptation [15, 40, 41]. In contrast, our virtual task required the same forces to move the hand with and without the objects. Indeed, participants expressed hand-specific adaptation despite participants not experiencing any perturbation during the Reach step of the Transport tasks. The two steps of the Transport task may have been poorly differentiated due to the absence of additional, non-visual cues.

In the Dual Learning experiment, participants could perform the object-transport movements differently when transporting objects with two distinct shapes matched to two distinct perturbations by the end of the Training phase. Using an altered process dissociation procedure, we attempted to qualify if these learning effects were due to the development of explicit strategies or implicit motor memories. We did not find conclusive evidence to suggest either subprocess of adaptation drove the overall learning effects. Studies exploring dual adaptation in conventional upper-limb motor learning tasks suggest a strong contribution of explicit processes [8, 42]. Although patterns of adaptation are similar in immersive virtual environments [39, 43–45], the effects of interactions within a new environments on the subprocesses of motor learning are not yet well explored. Early work suggests that adaptation in immersive environments may rely more heavily on explicit learning [43]. Further, when performing reaching tasks in immersive virtual environments, stable viewpoints while parallelly adapting to two perturbations, like in our study, may also lead to more explicit learning [39]. Furthermore, interacting in the virtual environment may have affected the outcome of the Dual Learning experiment due to the "newness" of the environment. Newly created internal models for new environmental contexts likely initially allow for high variability in predicted movements [1]. In immersive virtual environments where object interactions are unfamiliar to participants, new internal models of object transport may be flexible enough to encompass the variance of visual movements ascribable to opposing visuomotor perturbations. In future work, long-term exposure to virtual environments, or recruitment of participants with immersive-virtual-reality exposure, may mitigate such "newness" effects.

Considering people's ability to switch between different tools and the strong reliance on the visual system in motor adaptation, we predicted that visual shape properties of objects relevant to the performed movement could viably cue activating context-dependent motor memories. However, in this study, we find they are not sufficient for motor memory formation and switching in object-transport tasks. It is possible that in our tasks, the object-shape, although relevant to the task of choosing the target of the reach, was not sufficiently relevant to the lower-level task of transporting an object. That is, despite our task requiring explicit attention to the object-shape to determine the movement goal, it was not relevant to the execution of the movement. Additionally, motor adaptation, and interference from opposing perturbations, can often occur independently of explicit attention [46]. Visual properties of objects that affect the execution of the movement may sufficiently cue motor memory formation and switching. Conversely, it is possible that shape properties are simply not sufficient and different properties of tools, such as visual endpoints or tactile/proprioceptive factors, are needed to create and switch to object-specific motor memories. Overall, our findings suggest that movement-goal relevant object-shape cues may not be sufficient to create new context-dependent internal models for motor control when adapting to a single object-specific perturbation, and may act as weak but viable cues when adapting to multiple object-specific perturbations.

## Supporting information

**S1 Checklist.** *PLOS ONE* **clinical studies checklist.**
(DOCX)

## Acknowledgments

We thank the members of the Sensorimotor Control Laboratory for all their help.

## Author Contributions

**Conceptualization:** Shanaathanan Modchalingam, Maria N. Ayala, Denise Y. P. Henriques.

**Data curation:** Shanaathanan Modchalingam, Maria N. Ayala.

**Formal analysis:** Shanaathanan Modchalingam, Maria N. Ayala.

**Funding acquisition:** Shanaathanan Modchalingam, Denise Y. P. Henriques.

**Investigation:** Shanaathanan Modchalingam, Maria N. Ayala, Denise Y. P. Henriques.

**Methodology:** Shanaathanan Modchalingam, Maria N. Ayala, Denise Y. P. Henriques.

**Project administration:** Shanaathanan Modchalingam, Maria N. Ayala.

**Resources:** Denise Y. P. Henriques.

**Software:** Shanaathanan Modchalingam.

**Supervision:** Denise Y. P. Henriques.

**Validation:** Maria N. Ayala.

**Visualization:** Shanaathanan Modchalingam, Maria N. Ayala.

**Writing – original draft:** Shanaathanan Modchalingam, Maria N. Ayala.

**Writing – review & editing:** Shanaathanan Modchalingam, Maria N. Ayala, Denise Y. P. Henriques.

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
