## [Decision Letter · Decision Letter 0]

27 Jul 2023

PONE-D-23-13216Task-relevant object shape properties act as poor but viable cues for the attribution of motor errors to external objectsPLOS ONE

Dear Dr. Modchalingam,

Thank you for submitting your manuscript to PLOS ONE. After careful consideration, we feel that it has merit but does not fully meet PLOS ONE’s publication criteria as it currently stands. Therefore, we invite you to submit a revised version of the manuscript that addresses the points raised during the review process.

Your manuscript has been evaluated by three experts in the field. While reviewers consider your study interesting and well-conducted, they have also raised a few critical comments and concerns. In particular, there was a major concern regarding the appropriateness of the task design and the dependent variable used in this study.

We look forward to receiving your revised manuscript.

Kind regards,

Shenbing Kuang, Ph.D.

Academic Editor

PLOS ONE

"This work was supported by the Natural Science and Engineering Research Council of Canada Discovery Grant to DYPH and by the Natural Science and Engineering Research Council of Canada Post Graduate Scholarship - Doctoral to SM."

3. We note that Figure 1 in your submission contain copyrighted images. All PLOS content is published under the Creative Commons Attribution License (CC BY 4.0), which means that the manuscript, images, and Supporting Information files will be freely available online, and any third party is permitted to access, download, copy, distribute, and use these materials in any way, even commercially, with proper attribution. For more information, see our copyright guidelines: http://journals.plos.org/plosone/s/licenses-and-copyright.

Reviewers' comments:

Reviewer's Responses to Questions

**Comments to the Author**

1. Is the manuscript technically sound, and do the data support the conclusions?

Reviewer #1: Partly

Reviewer #2: No

Reviewer #3: Yes

2. Has the statistical analysis been performed appropriately and rigorously? 

Reviewer #1: No

Reviewer #2: Yes

Reviewer #3: Yes

3. Have the authors made all data underlying the findings in their manuscript fully available?

Reviewer #1: Yes

Reviewer #2: Yes

Reviewer #3: Yes

4. Is the manuscript presented in an intelligible fashion and written in standard English?

Reviewer #1: Yes

Reviewer #2: Yes

Reviewer #3: Yes

5. Review Comments to the Author

Reviewer #1: This study examined an object transporting task using the right arm in an immersive virtual-reality environment and examined the following two questions:

1) Is the extrinsic cue (object shape in this study) sufficient to allow for associated motor errors to be attributed to the object itself rather than the hand?

2) Does increasing task relevancy (associating two object shapes with different directions of visuomotor perturbation) allow for forming distinct motor memories for different extrinsic cues?

To test the first question, the authors conducted the Single Learning experiment. They asked whether learning of a 30-degree visuomotor rotation with a single object can be transferred to a new object with a different shape. The results showed that the learning effect fully transferred to the new object, suggesting that object shape is not a sufficient contextual cue to promote motor error assignment. On the other hand, the adaptation effect did not transfer sufficiently to the untrained hand regardless of the congruence or incongruence of the object shape. This suggests that intrinsic cues to movements, like used hands, may be helpful for context switching during motor learning.

To test the second question, they conducted the Dual Learning experiment. Two objects with different shapes were associated with two opposite visuomotor rotations to test if they could simultaneously learn the two distinct perturbations. The results showed that the subjects could not fully learn the two rotations. A small learning effect was partially observed, but due to the explicit strategy. Based on these results, the authors concluded that task-relevant visual properties of objects are insufficient for motor memory switching.

In motor neuroscience, the question of what information provides the context cues for motor memory switching is of great importance and has been extensively debated. As the authors mention, intrinsic cues to movements are known to be possible contextual information for the simultaneous formation of different motor memories. On the other hand, concerning the extrinsic cue on which this study focuses, it has been considered that it does not allow memory switching, except in some studies (e.g., Osu et al., 2004).

This study is novel in at least two respects. First, their task design gives higher task relevance to the extrinsic cue (object shape). The goal receptacle for the spherical object was a cylinder, and for the cubic object, a box. It has yet to be sufficiently investigated if increasing the task relevancy of such extrinsic cues may facilitate memory switching. Second, while most previous studies have used more conventional laboratory environments with robotic manipulandum, the authors tested the question under more realistic conditions in an immersive virtual-reality environment. Motor research using HMDs has been increasing in recent years, but the effects of virtual reality environments still need to be thoroughly investigated. Thus, this experimental data has a novelty and will be a valuable accumulation of knowledge for the field.

The introduction is well-written. The discussion of the results also makes sense. On the other hand, there are several issues regarding the description of the methodology and the robustness of the analysis and results. Specifically, I would like to mention the following points.

Major concerns

1) Methods: Please add a more detailed description of the experimental protocols. I think there was no description about how many trials each phase consisted of in each experiment.

2) Results: The medians of the three receptacle positions in the first and last trial-set were used in the statistical analysis. I recommend using the average, as the magnitude of the errors may depend on the receptacle position.

3) Results: I would like to know how robust the results of Experiment-2 are. The change in the hand angle thorough the experiment looks very small. The authors used only the first and last trial-set (three trials for each trial-set) to be analyzed. Please analyze using the first and last three trial-sets, as many studies often take the average of about ten trials.

4) Results: For Fig3D, the significant change in hand angle from baseline was observed only for the CCW rotation. The authors considered this change may be due to explicit (Fig. 3E). However, if the behavioral change during the Dual Learning experiment is due to an explicit process, why is there no change in CW? Also, the explicit process is believed to allow for quick and large behavioral changes. In contrast, why is the observed behavioral change so small if the explicit process works? The authors need to discuss these two points.

5) Result of Fig3D: Related to the fourth point, the authors mention that, although it may result from explicit strategy, the participants could utilize the object shape cues to change their movements depending on the context for the opposing visuomotor rotations. However, considering that only the CCW rotation led to a significant movement change in Fig. 3E, it is possible to think that it exhibited the learning for one context and the failure to learn for the other rather than showing different adaptive changes for the two contexts. This possibility should also be discussed.

Minor concerns

6) Methods: The receptacles were located 45, 90, and 180 degrees from the object's spawn location. The 45 and 180-degree target locations are asymmetric with respect to the 90-degree target, and only the 180-degree location is far from the other two locations. Please explain why these three positions were chosen.

7) Results: Please report whether or not there was any difference in the results between the three receptacle positions.

8) Discussion: In the last sentence (lines 449-450): "our findings suggest that task-relevant object-shape cues may not be sufficient to create new context-dependent internal models for motor control but rather update existing models.” The logic is unclear to me. What evidence from your results supports this argument? Could you please further explain on this point?

Reviewer #2: This paper examines if and how object shape might be used consolidate context-dependent motor plans. In the experimental protocol, an association is made between object shape/target (e.g. cube with square receptacle) and the deviation of the visually perceived movement trajectory in a virtual environment. Performance was evaluated using the “heading angle”, a measure based on the angular deviation from a straight line to the target at 3cm along the 10cm movement trajectory. This heading angle was used to represent initial directional error, prior to corrections based on feedback. Clamped trials where visually perceived position of the end effector along the movement trajectory was fixed (based upon displacement from departure point) were used to evaluate the post effect.

Based upon the results, movement adaptation did not appear to be retained when participants used their opposite hand during the clamped trials. At the same time, movement adaptation appeared present when the object was changed whilst using the same hand. The interpretation is that participants attribute errors to the hand being used rather than the object which is transported. This finding is a little curious as it seems that the hand was never affected during the reach stage (per results line 277 and 309), only during the transportation stage when the hand interacts with the object. This peculiarity does not seem to be addressed in the discussion.

Overall, there are many pleasing aspects of this study. Generally speaking, I found the paper well-written and easy to follow. The inclusion of Bayes factors for gauging the likelihood of the null hypotheses was very helpful. Despite this, I have a non-trivial concern regarding the methodological approach. The main variable used here (heading angle) is based on the angular deviation from a straight line early in the movement trajectory. It appears that the underlying assumption is that movement trajectories for this task should be straight. However, I don’t feel that this holds for the experimental conditions examined. In effect, the use of a cube shaped object with a square shaped receptacle implies a significant task constraint, in that the object must be aligned to the form of the receptacle upon placement. As a consequence, the movement trajectory will likely differ to that which would be used for point to point reaching. A person could exploit different postural configurations through the upper limb (shoulder, elbow, wrist, finger) in order to remain close to a linear pathway, although even this will be constrained by the position of the target in the peripersonal space. It is more likely that they would choose a curvilinear path with comfortable joint configuration (as we tend to do when we take a mug by the handle).

It appears that the authors attempted to make some adjustments for this issue. They report that individual postures according to target location were controlled for by subtracting HA angles at baseline from HA angles in the training (experimental) phase (see line 223). Still, I remain unconvinced by this approach. I feel that the best solution would have been examining the 6-DOF position of the rectangular object along with the upper limb kinematics (in particular the wrist). Of course, I would of course be much less concerned if the deposit zone was more simply a larger 2D drop off zone, implying much less stringent spatial constraints. The specific receptacle presented to subjects is highly specific, adjusted in dimensions (including depth) to accommodate for the visually perceived object. As it stands, the results indeed show statistically significant differences in heading angle between the objects with different shapes (p=0.0153, BF10 = 3.14; line 328). With these elements in mind, my concern is that the study design and dependent variable used in this study are questionable. Given this, I would not recommend accepting this paper.

Reviewer #3: This paper presented a study in which the authors examined the extent to which object shape cues can be used to support context-specific motor adaptation to visuomotor rotations. Overall, the design is sound and the presentation is mostly fine. I have a few concerns about the clarity of the experimental protocol and how the results should be interpreted.

Major

1. The authors argue that the object shape in this experiment is ‘task-relevant’. However, I think this may require more discussion. In my opinion, the object shape is task-relevant in a same way as object color, which is only associative to visuomotor rotation. But in every-day life, object shape could be more ‘task-relevant’ as it could provide information about task dynamics (e.g., weight, weigh distribution, functional goal, etc.). Therefore, while I agree with the findings, I am skeptical about the generalizability of the results to a broader consideration of what ‘object shape’ could provide as contextual cues. There were a few sentences in the last paragraph about this limitation, but I think this point should be emphasized to avoid misunderstanding of the results. I would also recommend the authors reconsider the use of ‘task-relevant’ in the title.

2. What does ‘exclude strategies’ mean exactly and how did the subjects interpret it?

Minor

1. It was not clear what the number of trials was for the experiments (fig. 1). I assume Training 1 had 180 trials, but how about Training 2, Test, and Aligned? Also, how many clamp trials? Please clarify.

6. PLOS authors have the option to publish the peer review history of their article (what does this mean?). If published, this will include your full peer review and any attached files.

Reviewer #1: No

Reviewer #2: No

Reviewer #3: No

---

## [Author Response · Author response to Decision Letter 0]

26 Jan 2024

We thank the editor and reviewers for their comments and recommendations. We address each comment separately below. 

In responses where we made changes to the manuscript, we indicated the relevant page and line numbers in the manuscript. For clarity, our responses are indicated in blue text in this document.

Thank you again for your time.

Shanaa Modchalingam,

Corresponding Author

General

https://journals.plos.org/plosone/s/file?id=ba62/PLOSOne_formatting_sample_title_authors_affiliations.pdf2. Thank you for stating in your Funding Statement: 

"This work was supported by the Natural Science and Engineering Research Council of Canada Discovery Grant to DYPH and by the Natural Science and Engineering Research Council of Canada Post Graduate Scholarship - Doctoral to SM."

New funding statement:

This work was supported by the Natural Science and Engineering Research Council of Canada (NSERC) Discovery Grant for DYPH, by the NSERC Post Graduate Scholarship - Doctoral for SM and MNA, by the NSERC-CREATE-IRTG Brain in Action Doctoral Scholarship for SM and MNA, and by the Vision: Science to Application Doctoral Entrance Award for SM.

There was no additional external funding received for this study.

3. We note that Figure 1 in your submission contain copyrighted images. All PLOS content is published under the Creative Commons Attribution License (CC BY 4.0), which means that the manuscript, images, and Supporting Information files will be freely available online, and any third party is permitted to access, download, copy, distribute, and use these materials in any way, even commercially, with proper attribution. For more information, see our copyright guidelines: http://journals.plos.org/plosone/s/licenses-and-copyright.

All images in Figure 1 were created by the corresponding author. All 3D renders (Figure 1 a and b) were created using Poser 11, and all plots and overlays were created using R and Inkscape respectively.

Reviewers' comments:

Reviewer's Responses to Questions

Comments to the Author

1. Is the manuscript technically sound, and do the data support the conclusions?

Reviewer #1: Partly

Reviewer #2: No

Reviewer #3: Yes

2. Has the statistical analysis been performed appropriately and rigorously?

Reviewer #1: No

Reviewer #2: Yes

Reviewer #3: Yes

3. Have the authors made all data underlying the findings in their manuscript fully available?

Reviewer #1: Yes

Reviewer #2: Yes

Reviewer #3: Yes

4. Is the manuscript presented in an intelligible fashion and written in standard English?

Reviewer #1: Yes

Reviewer #2: Yes

Reviewer #3: Yes

5. Review Comments to the Author

 

Reviewer 1

Reviewer #1: This study examined an object transporting task using the right arm in an immersive virtual-reality environment and examined the following two questions:

1) Is the extrinsic cue (object shape in this study) sufficient to allow for associated motor errors to be attributed to the object itself rather than the hand?

2) Does increasing task relevancy (associating two object shapes with different directions of visuomotor perturbation) allow for forming distinct motor memories for different extrinsic cues?

To test the first question, the authors conducted the Single Learning experiment. They asked whether learning of a 30-degree visuomotor rotation with a single object can be transferred to a new object with a different shape. The results showed that the learning effect fully transferred to the new object, suggesting that object shape is not a sufficient contextual cue to promote motor error assignment. On the other hand, the adaptation effect did not transfer sufficiently to the untrained hand regardless of the congruence or incongruence of the object shape. This suggests that intrinsic cues to movements, like used hands, may be helpful for context switching during motor learning.

To test the second question, they conducted the Dual Learning experiment. Two objects with different shapes were associated with two opposite visuomotor rotations to test if they could simultaneously learn the two distinct perturbations. The results showed that the subjects could not fully learn the two rotations. A small learning effect was partially observed, but due to the explicit strategy. Based on these results, the authors concluded that task-relevant visual properties of objects are insufficient for motor memory switching.

In motor neuroscience, the question of what information provides the context cues for motor memory switching is of great importance and has been extensively debated. As the authors mention, intrinsic cues to movements are known to be possible contextual information for the simultaneous formation of different motor memories. On the other hand, concerning the extrinsic cue on which this study focuses, it has been considered that it does not allow memory switching, except in some studies (e.g., Osu et al., 2004).

This study is novel in at least two respects. First, their task design gives higher task relevance to the extrinsic cue (object shape). The goal receptacle for the spherical object was a cylinder, and for the cubic object, a box. It has yet to be sufficiently investigated if increasing the task relevancy of such extrinsic cues may facilitate memory switching. Second, while most previous studies have used more conventional laboratory environments with robotic manipulandum, the authors tested the question under more realistic conditions in an immersive virtual-reality environment. Motor research using HMDs has been increasing in recent years, but the effects of virtual reality environments still need to be thoroughly investigated. Thus, this experimental data has a novelty and will be a valuable accumulation of knowledge for the field.

The introduction is well-written. The discussion of the results also makes sense. On the other hand, there are several issues regarding the description of the methodology and the robustness of the analysis and results. Specifically, I would like to mention the following points.

Major concerns

1) Methods: Please add a more detailed description of the experimental protocols. I think there was no description about how many trials each phase consisted of in each experiment.

Thank you, we have expanded the “Experiment protocol” section of the Methods substantially to add detail.

Page 9, Line 178.

2) Results: The medians of the three receptacle positions in the first and last trial-set were used in the statistical analysis. I recommend using the average, as the magnitude of the errors may depend on the receptacle position.

We thank the reviewer for the suggestion. We originally used medians to be less sensitive to outliers as each trial set included a small number of trials (3). For the reasons suggested by the reviewer, we now use the mean. The choice between means and medians of the trial-sets does not impact any of our findings. 

3) Results: I would like to know how robust the results of Experiment-2 are. The change in the hand angle thorough the experiment looks very small. The authors used only the first and last trial-set (three trials for each trial-set) to be analyzed. Please analyze using the first and last three trial-sets, as many studies often take the average of about ten trials.

We tend to use small set of trial for the initial set (one trial for each unique target) because given that learning curve can rise quite quickly, averaging across a large set of trials to repeated targets can greatly inflate the size of early compensation. This would be the case if we averaged the first 9 trials. 

However, there is no reason why we can’t use more trials for the Final set, given that learning should be quite stable by the end, so we use the last 3 trial-sets (9 trials) for the Final set. In this instance learning in the Dual-task is indeed unstable and small in the Dual Learning experiment. This change affected our conclusions about learning in the second experiment. We now do not find evidence of change in performance from the Initial to the Final set in the Training phase. Participants do however still reliably counter part of the rotation by the end of training (significantly different from Baseline performance: (t(29) = 3.68, p < 0.001, BF10 = 35.0)), suggesting a significant proportion of the observed learning occurs during the 3-trial Initial set.

We have changed the wording on page 16, line 309+ to account for the change.

4) Results: For Fig3D, the significant change in hand angle from baseline was observed only for the CCW rotation. The authors considered this change may be due to explicit (Fig. 3E). However, if the behavioral change during the Dual Learning experiment is due to an explicit process, why is there no change in CW? Also, the explicit process is believed to allow for quick and large behavioral changes. In contrast, why is the observed behavioral change so small if the explicit process works? The authors need to discuss these two points.

We thank the reviewer for the suggestion. Given the size of the effect as mentioned by the reviewer, we no longer think this is strong enough evidence for an explicit process. There is evidence for dual adaptation in the Dual Learning experiment, but it is unclear whether there is a driving effect of either an implicit or explicit process. 

We clarified this in the Results section: page 17, line 335.

5) Result of Fig3D: Related to the fourth point, the authors mention that, although it may result from explicit strategy, the participants could utilize the object shape cues to change their movements depending on the context for the opposing visuomotor rotations. However, considering that only the CCW rotation led to a significant movement change in Fig. 3E, it is possible to think that it exhibited the learning for one context and the failure to learn for the other rather than showing different adaptive changes for the two contexts. This possibility should also be discussed.

As with the explicit strategies mentioned in our reply above, we don’t think the results are strong enough (the effect is small and variable) to argue that learning, as measured during the clamp trials, occurred for one rotation or another. We feel confident that participants could differentiate behaviour between the two perturbations. 

We clarified this in the Results section: page 17, line 329.

Minor concerns

6) Methods: The receptacles were located 45, 90, and 180 degrees from the object's spawn location. The 45 and 180-degree target locations are asymmetric with respect to the 90-degree target, and only the 180-degree location is far from the other two locations. Please explain why these three positions were chosen.

This was a mistake in the draft, targets were equidistant from each other at 45, 90, and 135 degrees relative to the object spawn location. 

Fixed on page 8, line 148.

7) Results: Please report whether or not there was any difference in the results between the three receptacle positions.

We have added these calculations to a supplementary analysis notebook in the open-source repository. 

There were no effects of target position on the visuomotor adaptation results during the Training phase of either experiment. 

There was an effect of target position on the dual adaptation results during clamped trials (Figure 3d and e) but this effect did no interact with our findings (the effects of perturbation-direction). Hand angles were higher when transporting objects to the 45-degree target in all conditions.

The supplementary analysis can be found in the project repository (https://osf.io/twgc8/): “supplementary_analysis_nb.nb.html”.

8) Discussion: In the last sentence (lines 449-450): "our findings suggest that task-relevant object-shape cues may not be sufficient to create new context-dependent internal models for motor control but rather update existing models.” The logic is unclear to me. What evidence from your results supports this argument? Could you please further explain on this point?

We have removed the point about updating of existing models.

 

Reviewer 2:

Reviewer #2: This paper examines if and how object shape might be used consolidate context-dependent motor plans. In the experimental protocol, an association is made between object shape/target (e.g. cube with square receptacle) and the deviation of the visually perceived movement trajectory in a virtual environment. Performance was evaluated using the “heading angle”, a measure based on the angular deviation from a straight line to the target at 3cm along the 10cm movement trajectory. This heading angle was used to represent initial directional error, prior to corrections based on feedback. Clamped trials where visually perceived position of the end effector along the movement trajectory was fixed (based upon displacement from departure point) were used to evaluate the post effect.

Based upon the results, movement adaptation did not appear to be retained when participants used their opposite hand during the clamped trials. At the same time, movement adaptation appeared present when the object was changed whilst using the same hand. The interpretation is that participants attribute errors to the hand being used rather than the object which is transported. This finding is a little curious as it seems that the hand was never affected during the reach stage (per results line 277 and 309), only during the transportation stage when the hand interacts with the object. This peculiarity does not seem to be addressed in the discussion.

We added this to the discussion on Page 21, Line 420+ when comparing our task to real-world object transport tasks. We think the lack of non-visual cues (e.g. tactile and weight cues) that occur in real-world object-transport tasks was lacking in our virtual environment. Since movement dynamics were the same when transporting objects, the transport step may have been interpreted as a pseudo-reach.

Overall, there are many pleasing aspects of this study. Generally speaking, I found the paper well-written and easy to follow. The inclusion of Bayes factors for gauging the likelihood of the null hypotheses was very helpful. Despite this, I have a non-trivial concern regarding the methodological approach. The main variable used here (heading angle) is based on the angular deviation from a straight line early in the movement trajectory. It appears that the underlying assumption is that movement trajectories for this task should be straight. However, I don’t feel that this holds for the experimental conditions examined. In effect, the use of a cube shaped object with a square shaped receptacle implies a significant task constraint, in that the object must be aligned to the form of the receptacle upon placement. As a consequence, the movement trajectory will likely differ to that which would be used for point to point reaching. A person could exploit different postural configurations through the upper limb (shoulder, elbow, wrist, finger) in order to remain close to a linear pathway, although even this will be constrained by the position of the target in the peripersonal space. It is more likely that they would choose a curvilinear path with comfortable joint configuration (as we tend to do when we take a mug by the handle).

Thank you for this great point. Firstly, we ensured that participants did not need to orient the cube object to match the orientation of the receptacle when transporting to the target. This was made clear on the practice trials. We clarified this in the Methods section (page 7, lines 126). 

Additionally, we conducted supplementary analyses to ensure participants do not differ in hand-position when initiating the Transport step of the task with different objects. The supplementary analysis can be found in the project repository (https://osf.io/twgc8/): “supplementary_analysis_nb.nb.html”.

It may be that participants used a slightly curvilinear path to move to certain targets, but the results from our Single Learning experiment suggest that participants do deviate the initial direction of the hand movement in the expected direction by 23.7 degrees. In order to achieve the target during these reaches, hand paths are likely slightly curved due to feedback-based corrections, but this is expected behaviour even in point-to-point reaching tasks. 

It appears that the authors attempted to make some adjustments for this issue. They report that individual postures according to target location were controlled for by subtracting HA angles at baseline from HA angles in the training (experimental) phase (see line 223). Still, I remain unconvinced by this approach. I feel that the best solution would have been examining the 6-DOF position of the rectangular object along with the upper limb kinematics (in particular the wrist). Of course, I would of course be much less concerned if the deposit zone was more simply a larger 2D drop off zone, implying much less stringent spatial constraints. The specific receptacle presented to subjects is highly specific, adjusted in dimensions (including depth) to accommodate for the visually perceived object. As it stands, the results indeed show statistically significant differences in heading angle between the objects with different shapes (p=0.0153, BF10 = 3.14; line 328). With these elements in mind, my concern is that the study design and dependent variable used in this study are questionable. Given this, I would not recommend accepting this paper.

Thank you for this suggestion. The receptacles openings were indeed larger than the objects being transported (2x the size of the objects) for this reason – twice the size of the transported object. Participants were not required to orient the objects to the receptacle and received practice trials to learn such nuances of the task. This was not clear in the original draft of the manuscript, and we have clarified. 

Page 7, line 124.

 

Reviewer 3:

Reviewer #3: This paper presented a study in which the authors examined the extent to which object shape cues can be used to support context-specific motor adaptation to visuomotor rotations. Overall, the design is sound and the presentation is mostly fine. I have a few concerns about the clarity of the experimental protocol and how the results should be interpreted.

Major

1. The authors argue that the object shape in this experiment is ‘task-relevant’. However, I think this may require more discussion. In my opinion, the object shape is task-relevant in a same way as object color, which is only associative to visuomotor rotation. But in every-day life, object shape could be more ‘task-relevant’ as it could provide information about task dynamics (e.g., weight, weigh distribution, functional goal, etc.). Therefore, while I agree with the findings, I am skeptical about the generalizability of the results to a broader consideration of what ‘object shape’ could provide as contextual cues. There were a few sentences in the last paragraph about this limitation, but I think this point should be emphasized to avoid misunderstanding of the results. I would also recommend the authors reconsider the use of ‘task-relevant’ in the title.

We thank the reviewer for this feedback. We agree that task-relevant may not be the right terminology. The object shapes were relevant to the end-goal of the movements, whereas extrinsic cues like colour in past studies were not (that is, the movement task could be completed even while ignoring the colour cue). We have made changes throughout the Introduction and Discussion section of the manuscript to use the new “goal relevant” terminology.

We have also changed the title to accommodate the new terminology.

2. What does ‘exclude strategies’ mean exactly and how did the subjects interpret it?

We used an altered form of the Process Dissociation Procedure as used by Werner at al. (2015). We have clarified this in the Methods section (Page 11, Line 217+) and added relevant citations to direct the reader to more details.

Werner, S., Van Aken, B. C., Hulst, T., Frens, M. A., Van Der Geest, J. N., Strüder, H. K., & Donchin, O. (2015). Awareness of sensorimotor adaptation to visual rotations of different size. PLoS ONE, 10(4). https://doi.org/10.1371/journal.pone.0123321

Minor

1. It was not clear what the number of trials was for the experiments (fig. 1). I assume Training 1 had 180 trials, but how about Training 2, Test, and Aligned? Also, how many clamp trials? Please clarify.

Thank you for the suggestion. Due to this and Review 1’s comments, we have expanded the “Experiment protocol” section of the Methods substantially to add detail.

Page 9, Line 178. 

6. PLOS authors have the option to publish the peer review history of their article (what does this mean?). If published, this will include your full peer review and any attached files.

Do you want your identity to be public for this peer review? For information about this choice, including consent withdrawal, please see our Privacy Policy.

Reviewer #1: No

Reviewer #2: No

Reviewer #3: No

---

## [Decision Letter · Decision Letter 1]

21 Feb 2024

Movement-goal relevant object shape properties act as poor but viable cues for the attribution of motor errors to external objects

PONE-D-23-13216R1

Dear Dr. Modchalingam,

We’re pleased to inform you that your manuscript has been judged scientifically suitable for publication and will be formally accepted for publication once it meets all outstanding technical requirements.

Kind regards,

Shenbing Kuang, Ph.D.

Academic Editor

PLOS ONE

Additional Editor Comments (optional):

Reviewers' comments:

Reviewer's Responses to Questions

**Comments to the Author**

1. If the authors have adequately addressed your comments raised in a previous round of review and you feel that this manuscript is now acceptable for publication, you may indicate that here to bypass the “Comments to the Author” section, enter your conflict of interest statement in the “Confidential to Editor” section, and submit your "Accept" recommendation.

Reviewer #1: (No Response)

Reviewer #3: All comments have been addressed

2. Is the manuscript technically sound, and do the data support the conclusions?

Reviewer #1: Partly

Reviewer #3: Yes

3. Has the statistical analysis been performed appropriately and rigorously? 

Reviewer #1: Yes

Reviewer #3: Yes

4. Have the authors made all data underlying the findings in their manuscript fully available?

Reviewer #1: Yes

Reviewer #3: Yes

5. Is the manuscript presented in an intelligible fashion and written in standard English?

Reviewer #1: Yes

Reviewer #3: Yes

6. Review Comments to the Author

Reviewer #1: I thank the authors for the revised version of the manuscript. I found responses to previous concerns satisfactory. I have one last concern regarding Fig 3 for the Dual learning experiment.

In Fig 3A, as described in Methods, the authors used normalized HAs to align the direction of the HA change to two opposing visuomotor rotations. A change in the positive direction indicates that the HA changed to compensate for the rotation in each condition. In Fig 3C and D, I am wondering if the authors are using normalized HAs? If so, negative values in CCW indicate that learning performance got worse. This is relevant to the conclusion, so please be sure to provide an explanation for this in the text.

I would also like to see what the learning curve looks like using non-normalized HAs for each of the opposing rotations, and I would suggest that the authors present the figure in supplementary results.

Reviewer #3: (No Response)

7. PLOS authors have the option to publish the peer review history of their article (what does this mean?). If published, this will include your full peer review and any attached files.

Reviewer #1: No

Reviewer #3: No

---

## [Editor Report · Acceptance letter]

19 Mar 2024

PONE-D-23-13216R1 

PLOS ONE

Dear Dr. Modchalingam, 

I'm pleased to inform you that your manuscript has been deemed suitable for publication in PLOS ONE. Congratulations! Your manuscript is now being handed over to our production team.

Kind regards, 

on behalf of

Associate Professor Shenbing Kuang 

Academic Editor

PLOS ONE